# Upper Limb Movement Measurement Systems for Cerebral Palsy: A Systematic Literature Review

**DOI:** 10.3390/s21237884

**Published:** 2021-11-26

**Authors:** Celia Francisco-Martínez, Juan Prado-Olivarez, José A. Padilla-Medina, Javier Díaz-Carmona, Francisco J. Pérez-Pinal, Alejandro I. Barranco-Gutiérrez, Juan J. Martínez-Nolasco

**Affiliations:** 1Electronics Engineering Department, National Technology of Mexico in Celaya, Celaya 38010, Mexico; d1903003@itcelaya.edu.mx (C.F.-M.); alfredo.padilla@itcelaya.edu.mx (J.A.P.-M.); javier.diaz@itcelaya.edu.mx (J.D.-C.); francisco.perez@itcelaya.edu.mx (F.J.P.-P.); israel.barranco@itcelaya.edu.mx (A.I.B.-G.); 2Mechatronics Engineering Department, National Technology of Mexico in Celaya, Celaya 38010, Mexico; juan.martinez@itcelaya.edu.mx

**Keywords:** Kinect, kinematics, active range of motion, measurement, upper limb, cerebral palsy

## Abstract

Quantifying the quality of upper limb movements is fundamental to the therapeutic process of patients with cerebral palsy (CP). Several clinical methods are currently available to assess the upper limb range of motion (ROM) in children with CP. This paper focuses on identifying and describing available techniques for the quantitative assessment of the upper limb active range of motion (AROM) and kinematics in children with CP. Following the screening and exclusion of articles that did not meet the selection criteria, we analyzed 14 studies involving objective upper extremity assessments of the AROM and kinematics using optoelectronic devices, wearable sensors, and low-cost Kinect sensors in children with CP aged 4–18 years. An increase in the motor function of the upper extremity and an improvement in most of the daily tasks reviewed were reported. In the population of this study, the potential of wearable sensors and the Kinect sensor natural user interface as complementary devices for the quantitative evaluation of the upper extremity was evident. The Kinect sensor is a clinical assessment tool with a unique markerless motion capture system. Few authors had described the kinematic models and algorithms used to estimate their kinematic analysis in detail. However, the kinematic models in these studies varied from 4 to 10 segments. In addition, few authors had followed the joint assessment recommendations proposed by the International Society of Biomechanics (ISB). This review showed that three-dimensional analysis systems were used primarily for monitoring and evaluating spatiotemporal variables and kinematic parameters of upper limb movements. The results indicated that optoelectronic devices were the most commonly used systems. The joint assessment recommendations proposed by the ISB should be used because they are approved standards for human kinematic assessments. This review was registered in the PROSPERO database (CRD42021257211).

## 1. Introduction

Kinematics is the study of motion without attending to the forces that create the movement. In this sense, the kinematics of the human body describes the motion, type, direction, magnitude, location in space, and rate of change of velocity of its bony segments [1].

A kinematic analysis describes the biomechanical characteristics of motor function and provides objective and accurate information [2]. However, the accuracy is dependent on the measuring instrument used. This type of measurement is relevant for a better understanding of any risks or deficiencies involving specific parts of the human body. It contributes to clinical decision-making concerning the need and type of surgery, drug administration, and even the assessment of treatment efficacy [3]. In clinical settings, a post-stroke recovery analysis is a recommended application [4]. However, a kinematic analysis is also used to assess the upper limb movement performance of patients with clinical conditions such as CP [5]. CP is associated with a set of disorders that cause movement and postural impairments, resulting in limitations in daily functional activities [6]. According to the World Health Organization statistics, it is estimated that more than one billion individuals have a physical limitation, comprising approximately 15% of the world’s population [7]. CP is the most recurrent and expensive cause of motor limitation in infancy [8]. Its frequency is 2.4 cases per 1000 infants born; in very pre-term infants, this figure increases to 40–100 cases per 1000 [9]. An early diagnosis can significantly help when selecting an adequate rehabilitation technique and improves the rehabilitation process of the patient [10]. An efficient, objective, and accurate measurement of the upper limb motor performance is relevant primarily to monitor the use of the upper limbs in the daily activities of people with CP and to improve the patient’s rehabilitation treatments. Clinical ROM measurements are essential for monitoring the progress of motor rehabilitation as these assess the degree of impairment and evolution of the patient [1]. In general terms, there are two ways to measure a ROM: the passive range of motion (PROM) and the AROM. A PROM analysis involves a method of movement assisted by a person or by a mechanical device. In this paper, we considered presenting the techniques of a self-assisted assessment of motor skills. For this reason, this review focuses on the former. The AROM of the upper limbs is an essential issue in a patient’s rehabilitation process because it provides objective and helpful information in the diagnosis and clinical follow-up of the physical condition progress.

The Melbourne Assessment of Unilateral Upper Limb Function (MUUL), Quality of Upper Extremity Skills Test (QUEST), and Shriners Hospital Upper Extremity Evaluation (SHUEE) are methods that qualitatively determine the motor performance of the upper extremity whilst performing various functional tasks [11]. However, the application of these methods only provides subjective results of limb motor performance as they employ weighted scores based on the evaluator’s experience and observations, which is not always accurate. As Bard et al. [12] point out, to measure the efficacy or consequences of a new treatment in children with CP, it is essential to use valid and reliable quantitative tools; they affirm that a subjective observation by a specialist is not enough. Meanwhile, the application of quantitative measurement scales provides an objective result of the upper limb performance based on measurements and calculations of joint angles, movement duration, and velocity. De los Reyes-Guzmán et al. [13] proposed several kinematic metrics of quantitative evaluation related to the upper limb movement such as speed, efficacy, efficiency, accuracy, smoothness, control strategy (time to velocity peak), and the functional ROM. These metrics can be applied in conjunction with clinical scales for the functional assessment of the upper extremity in patients with neurological injuries such as CP as stated by Jaspers et al. [5] who addressed quantitative measurement systems for analyzing reach, gross, and fine motor functions as well as spatiotemporal characteristics. Furthermore, they stated that it is necessary to complement this type of objective study with qualitative scales for a better assessment of the upper extremity, which is indispensable for selecting the best treatment of a patient with CP; the devices reported at that time ranged from electrogoniometers to optoelectronic systems.

Traditionally, the ROM is measured using a universal goniometer because it is an inexpensive, transportable, and easy to use device [14]. Other instruments currently used are electrogoniometers [15], laser goniometers [16], digital inclinometers [17], optoelectronic devices [18,19,20,21], smartphone applications [22,23], wearable sensors [24,25,26,27], and low-cost sensors [28] including Kinect sensors [29,30]. It is also necessary to mention that Xsens systems allow the determination of the ROM and kinematic variables. For example, authors Franco et al. [31] used an Xsens device (Xsens, Enschede, Netherlands) to obtain kinematic data of the shoulder in healthy people during different loading conditions. They have also been applied in the field of ergonomics as well as lower limb assessments [31,32].

Recently, researchers have shown an increased interest in the use of Kinect sensors in health-related areas [33,34,35]. Other applications of this device are in the area of agronomy, education, cartography, and robotics [36,37]. Studies utilizing virtual reality (VR) systems for improving the functional skills of the upper limbs have shown the Kinect platform to be a promising rehabilitation tool for CP treatments [38,39,40,41,42,43,44,45,46,47,48,49]. In this context, Microsoft Kinect v2 (Microsoft, Redmond, WA, USA) is an RGB-D depth sensor that uses a grid of infrared light. It consists of two cameras: an RGB camera and an infrared camera. Following the time-of-flight principle, the software development kit generates a list of coordinates (X, Y, Z) as a point cloud generating an artificial skeleton based on 25 artificial anatomical markers projected onto depth data [50]. Similarly, a three-dimensional (3D) motion study is a tool that allows an objective and quantitative evaluation of the ROM in all degrees of freedom [5,51].

Currently, it is necessary to know how new technologies support the objective assessment of the upper limbs in patients with CP. In this study, we assessed quantitative measurement systems from optoelectronic devices and wearable sensors to new 3D measurement technologies with a natural user interface (NUI), commercially known as Kinect (v1/v2).

This article aims to review the current status of available techniques for the quantitative estimation of the AROM and kinematics of the upper limbs of patients with motor impairments due to CP.

## 2. Methods

This paper is a bibliographic review regarding the quantitative assessment of the upper extremity in patients with CP through optoelectronic devices, wearable sensors, and Kinect. The concept of the quantitative assessment refers to the AROM and kinematic assessment of the affected upper extremity in patients with CP.

### 2.1. Eligibility Criteria

In this study, the inclusion criteria were all study designs that included objective upper extremity assessments of both the AROM and kinematics in children with CP aged 4–18 years using optoelectronic devices, wearable sensors, and low-cost Kinect sensors, which also included qualitative upper extremity assessments with clinically validated scales.

### 2.2. Search Strategies

Papers published between 2011 and March 2021 were reviewed. Without language restrictions, the main databases (Science Direct, PubMed, Cochrane Library, EBSCO host, IEEE, Google Scholar) and specialized journals such as Developmental Medicine & Child Neurology and Frontiers in Neurology were searched. Different search terms including “cerebral palsy”, “active movement”, “movement analysis”, “upper limb”, “kinematics”, “quantitative assessment”, “goniometer”, “Kinect”, “low-cost device”, “natural user interface”, “RGB-D sensor”, “virtual reality”, “precision”, “accuracy”, “reliability”, and “goniometer laser” were applied in the search for articles relevant to the research topic. This review was registered in the PROSPERO database (CRD42021257211).

### 2.3. Study Selection

Figure 1 shows the selection of studies with the articles screened and the selection process according to the PRISMA format. Several studies performed quantitative assessments of cases other than CP during the data extraction; therefore, 32 studies were excluded. In addition, we found assessments involving non-child patients and healthy subjects; therefore, we discarded 101 such studies. Similarly, after a consensus among the authors, we decided to include two studies that did not mention any clinical scale. However, these studies provided valuable information for the quantitative assessment of this study population. Finally, 14 primary papers were selected for the analysis. The references of the selected studies were grouped and formatted using the EndNote (Clarivate Analytics, London, UK) reference manager.

### 2.4. Data Extraction

To improve the integrity, transparency, quality, and consistency of the research, the PRISMA methodology was applied [52]. Similarly, the methodological quality of the selected studies was scored using the Downs and Black scale [53]. In this study, we did not consider the power section score as this was not available in most of the studies analyzed; therefore, the maximum score was 27, similar to a prior classification [54]; a score of 25–27 was measured as excellent, 19–24 was good, 14–18 was fair, and less than or equal to 13 was poor. All studies were included regardless of the methodological quality levels in this review. However, in most studies, the methodological quality ranged from good to excellent.

The number of patients who participated in the study, types of intervention, parameters measured, assessment instrument, clinical scale, and study results were extracted from the selected articles. Table 1 summarizes the main characteristics of the study population and the scales applied in the assessment of movement quality.

## 3. Current Techniques for Objective Measurements

Motion capture systems based on optoelectronic devices (Vicon systems) (Oxford Metrics Group, UK) are considered to be the gold standard for a motion analysis. However, the analysis requires sending patients to a clinical laboratory and is most commonly used to assess kinematics and spatiotemporal variables during activities such as reaching and grasping [55,56,57,59,62,64,66]. Wearable inertial sensors also directly measure kinematic data of the human body and are an alternative able to be used both in the laboratory and in an open environment [61,63,67]. Another less expensive alternative with more advantages is the use of Kinect devices (Microsoft, Redmond, WA, USA) [60,65]. The frequency of using quantitative evaluation methods for both the AROM and kinematics in primary studies is presented in Figure 2.

### 3.1. ROM Assessment

Table 2 shows the main characteristics of the quantitative evaluation systems included in this study.

As can be seen from Table 2, OptiTrack (Motion Analysis, Corvallis, OR, USA), ELITE2002 (BTS, Milan, Italy), and Vicon Mx optoelectronic devices (Oxford Metrics Group, UK) were used to record significant changes (*p* < 0.05) regarding the ROM of abduction, external rotation, flexion, extension, supination, and other movements during the respective evaluations according to the applied protocol [51,55,56,57,58,59,62,64].

Low-cost Kinect sensors (Microsoft, Redmond, WA, USA) were mainly used to acquire angular data and quantify the ROM for the movements of flexion, extension, abduction, internal rotation, and external rotation of the upper limbs [60,65]. Daoud et al. [65] indicated that the variability of the values obtained for shoulder flexion, abduction, and adduction movements in a group of six children depended on the degree of cerebral palsy involvement. The accuracy ranged from 83% to 85%, the specificity from 83% to 86%, and the sensitivity from 83% to 84%. In one study [60], one participant’s pre-intervention radial/ulnar wrist movements ranged from 29° to 36°. After treatment with the aid of a Kinect v1 and FAAST software, this participant showed an improvement of 15° to 27° relative to that on the unaffected side, which ranged from 10° to 15°.

The Kinect v2 sensor (Microsoft, Redmond, WA, USA) is not only low-cost but also effective compared with optoelectronic devices [51,55,56,57,58,59,62,64] as it has a color camera with a resolution of 1920 × 1080 pixels at 30 fps and a depth camera with a resolution of 512 × 424 at 30 fps, with a field of view of 70° horizontal and 60° vertical. It also allows the determination of 25 joint points without the need for skin markers, unlike Vicon Mx optoelectronic devices (Oxford Metrics Group, UK) or OptiTrack (Motion Analysis, Corvallis, OR, USA) [65,68].

Optoelectronic devices, low-cost Kinect sensors, and inertial sensors have all been used to quantify the upper extremity ROM in clinical cases of CP [67]. In [67], with the use of an inertial sensor, the range of flexion movement with a dynamic elbow deformity was determined to vary from 35.4° to −46.5°.

### 3.2. Kinematic Analysis

Table 3 shows the main spatiotemporal parameters obtained with optoelectronic devices and portable sensors that have allowed the determination of significant differences between the affected and unaffected limbs of patients with CP.

As Table 3 shows, optoelectronic devices were also used to determine spatiotemporal parameters such as velocity and acceleration during the activities of each assessment [57,62,66]. Meanwhile, several authors used inertial sensors to simultaneously obtain the angular velocity and acceleration of their patients [61,63]. Other authors used Vicon devices to determine the smoothness of the movements of the activities performed during the evaluation [64] and one [61] used an inertial sensor to measure the smoothness of the motion. The range of normality of the motion of each sub-movement performed in the test was calculated using the spectral arc length metric (SALM), peak metric (PM), and log dimensionless jerk metric (LDJM). One of the advantages of this device is that it performs measurements without reference points and can also operate in open spaces. However, the device still has to be placed on the body to obtain data on angular velocity, acceleration, or smoothness of motion.

## 4. Discussion

Several reports have shown that there are currently different tools to clinically assess the upper limb movement function in patients with CP [10,11]. However, these techniques are evaluator dependent, which makes the quantification of the upper limb motor function difficult.

### 4.1. Discussion of Assessment Methods

Several methods for a range of motion assessments are available; each is reliable when comparing results with a universal goniometer [19,20,22,69]. A 3D analysis using optoelectronic devices with reflective markers, video data capture, and an analysis using computational algorithms is currently the most accurate technique for assessing the ROM [70]. Few studies have reported the application of a Kinect sensor (Microsoft, Redmond, WA, USA) as a tool for the AROM and kinematic assessment of children with CP. However, after reviewing the scientific evidence available to date, a low-cost Kinect sensor presents more advantages for use than a device employing optoelectronics through 3D tracking markers as the skin markers may change position on the skin whilst assessing motion, thus increasing inaccuracy when mapping joint angles [71] (Table 4). Unfortunately, Kinect devices (Microsoft, Redmond, WA, USA) were discontinued in 2017 [68]. However, they are still in circulation and are still in use today [72]; this reflects a disadvantage as although they have a great potential in healthcare, it may not be possible to use this type of technology in the future. The features of Azure Kinect (Microsoft, Redmond, WA, USA) can overcome this drawback. The accuracy and reliability of this device were compared with Xsens (Xsens, Enschede, The Netherlands) and OptiTrack optoelectronic systems (Motion Analysis, Corvallis, OR, USA) in lower limb movements with minimal differences between the three devices [32].

### 4.2. Discussion of Optoelectronic Devices

As mentioned in the literature review, the joint application of kinematic and spatiotemporal parameters may allow for the identification of clinical movement patterns observed in patients with CP and, consequently, enable the therapist to plan an optimal upper limb treatment [56]. Howcroft et al. [57] noted that the variability of upper limb kinematic patterns should be investigated individually. Gaillard et al. [51] concluded that the assessment protocol they developed was a novel and practical tool to analyze abnormal upper limb movement patterns efficiently.

Previous studies have demonstrated that quantitative measurement techniques for a 3D movement analysis of the upper extremity are mainly related to optoelectronic systems. To date, Vicon systems (Oxford Metrics Group, UK) are still used more frequently [5]. However, this type of assessment system is not always accessible to all rehabilitation centers, mainly because of cost. Furthermore, as shown in Table 4, these devices have an accuracy of < 1 mm under optimal measurement conditions. This accuracy is principally affected by illumination, electromagnetic interference, the correct installation of the cameras, and procedures related to the placement of skin markers, which is why the motion analysis is performed in specialized laboratories with highly trained personnel, as mentioned in [73]. Scano et al. [74] noted that the Kinect could replace the Vicon system when it is not possible to use this technology or when the assessment is in an industrial setting. Their study demonstrated similarities in the accuracy of both devices at various angles.

### 4.3. Discussion of Wearable Sensors

Concerning the advantages of the wearable sensors shown in Table 4**,** it is relevant to note that both IMU devices and smartphone applications are integrated with similar components (accelerometer, gyroscope) and this is reflected in the accuracy of the device: 6.8 ± 2.7° with an ICC of 0.930–0.979 for IMUs and ± 3.6° with an ICC of 0.63–0.68 for smartphone apps. However, these devices have to be in direct contact with the patient. Currently, this would represent a risk in the face of the COVID-19 pandemic, as stated by Moreira et al. [75]; these assessment devices would not be a favorable option in this situation.

The results of Anaya et al. [61] indicated that the smoothness metric is an efficient measure of the upper limb movement that can be used to determine the differences between an affected and an unaffected limb. They are currently investigating the correlation of the results obtained with clinical scales and are developing an objective scale for a functional assessment based on SALM smoothness. Kim et al. [63] pointed out that dynamic motion measurements can be performed under normal conditions; that is, outside the laboratory in real-time using an accelerometer because of its portability.

Speed, efficacy, efficiency, accuracy, smoothness, control strategy (time to velocity peak), and the ROM are kinematic metrics of quantitative evaluations related to upper limb movements [13]. The results of this review have several relevant applications for future practice because the application of quantitative measurement scales provides an objective result of the upper limb performance.

### 4.4. Discussion of the Low-Cost Sensor, Kinect

As can be seen in Table 4, the Kinect sensor (Microsoft, Redmond, WA, USA) presented errors in the measurement of the sagittal plane as well as in the ulnar and radial deviation of the upper limbs. Similarly, Wasenmüller and Stricker [76] mention that factors such as temperature, distance, scene color, random noise, and flying pixels can affect the accuracy of the depth camera. They recommended that the Kinect should be pre-heated for 25 min to compensate for temperature, bilateral filtering, and other recommendations. It is relevant to consider and control all these variables when making measurements in the laboratory or another environment to obtain reliable measurements. In this context, Sevick et al. [60] argue that an intervention using a Kinect sensor could be successfully delivered in the laboratory and at home. Daoud et al. [65] claimed that the computerized assessment method provided kinematic measurement indicators to objectively measure movements such as smoothness and range. These authors suggested the application of this method as an additional technique for monitoring CP rehabilitation treatments.

When considering the fusion of a Kinect sensor with other sensors such as IMUs, they are a viable way to compensate for the accuracy of both devices in upper limb movements, as realized in [77]. Based on the above, the NUI of a Kinect sensor can be considered to be a VR technique [33] that obtains the ROM and kinematics of the upper limbs in patients with CP because it allows the evaluation of body movement, scene modeling, gesture recognition, rehabilitation, and posture reconstruction using vector data [46].

These findings recommend, in general, the inclusion of high-quality experimental multicenter studies and appropriate follow-up rehabilitation programs. An increase in the motor function of the upper extremity and an improvement in the daily tasks of most of the tasks reviewed were reported.

This review has several limitations such as limited access to other major bibliographic databases, e.g., EMBASE, CINAHL, and Scopus. This limitation to access information may cause a lack of relevant references in this review. Few authors had described in detail the kinematic models and algorithms used to estimate their kinematic analysis. In addition, few authors had followed the joint assessment recommendations proposed by the ISB. It is suggested to make use of this recommendation as it is an approved standard for human kinematic assessments.

## 5. Conclusions

This review has shown that 3D analysis systems are used primarily for monitoring and evaluating the spatiotemporal variables and kinematic parameters of upper limb movements. The results indicated that optoelectronic devices are the most commonly used systems. One clinical assessment tool with a unique advantage is the use of a Kinect sensor as it has a markerless motion capture system. Nevertheless, few studies have reported using a Kinect sensor as a tool for the AROM and kinematic assessment of the upper extremity in this group study. In addition, the Kinect device has been discontinued, reducing the likelihood of future use. However, a few online shops such as Amazon still have this device in stock. There is now the Azure Kinect sensor with enhanced features that can be used for this type of assessment.

In this review, the study population ranged from 4 to 50 patients, showing a higher frequency above 15 patients; the activities differed according to the protocols established in each study. However, the importance lies in each of these parameters measured with the various instruments because they indicated an increase in each movement performed with the affected upper limb. This variation of the sample, protocols, and evaluation devices did not allow for a homogenized result. Nevertheless, this review has shown an overview of optoelectronic devices, portable sensors, and mainly how low-cost Kinect sensors have been used as complementary tools for clinical evaluation scales. Although each parameter and device are different, for this reason, the specialist physician in this type of upper limb movement analysis must choose the instrument that best suits the needs of the protocol and rehabilitation of the patient whilst also considering the accessibility of these evaluation devices and the guidelines established by the ISB.

Finally, the potential of wearable sensors and Kinect as complementary devices for the quantitative evaluation of the upper extremity in this study population was evident. Further studies with wearable sensors and Kinect v2 sensors (available) or Azure Kinect sensors are strongly recommended. Larger samples in real-life settings should also be considered but under optimal measurement conditions to avoid errors.

## Figures and Tables

**Figure 1 sensors-21-07884-f001:**
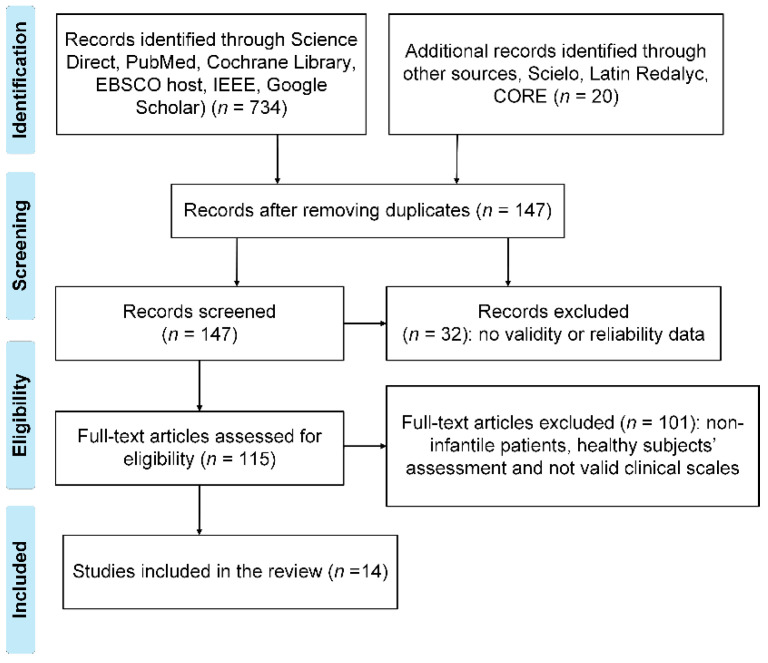
Flowchart showing the selection of the reviewed papers.

**Figure 2 sensors-21-07884-f002:**
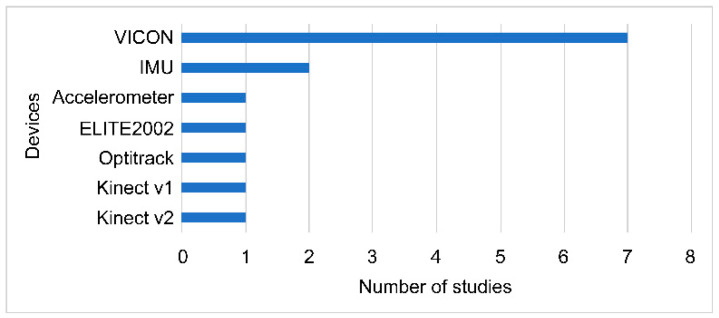
Frequencies used in the quantitative assessment systems.

**Table 1 sensors-21-07884-t001:** Population characteristics.

Author	Downs and Black (*n* = 27)	Population (*n*)	Protocol	Clinical Scale
Gaillard et al. [51]	22	20	Carry out four bimanual tasks during three consecutive evaluations	AHA and ABILHAND-Kids
Fitoussi et al. [55]	25	27	Two daily tasks named “to drink” and “to move an object”	Ashworth Scale
Jaspers et al. [56]	20	20	Three reach tasks (forward, upward, and sideways) for two sessions	MACS
Howcroft et al. [57]	22	17	Play four games for 8 min with a 5 min rest interval in between	Physical Activity Enjoyment Scale (PACES)
Galli et al. [58]	23	16	3D video recording and gait analysis along a 10 m walkway	ROM
Sarcher et al. [59]	22	15	Perform eight different tasks for four cycles with a 2 min break between each task	Modified MACS and Ashworth Scale (MAS)
Sevick et al. [60]	19	4	Playing four different games in an hour three times a week over twelve weeks	GMFCS, MACS, and BOT-2 Bruininks–Oseretsy Motor Behavior Test Scale
Anaya Campos et al. [61]	19	16	Insert pieces into a board three times on each side and then return them to their original places	MAS, Tardieu, MACS, and BFMF
Mailleux et al. [62]	24	50	Eight movement repetitions per task	MACS, AHA, and Melbourne Assessment 2
Kim et al. [63]	20	15	Perform a scoping task three times with a 10 s break between each assessment	Jebsen Taylor Hand Function Test (JTHFT), QUEST, Box and Blocks Test (BBT), and ABILHAND-Kids
Cacioppo et al. [64]	23	20	Perform five bimanual tasks during a complete cycle	MACS
Daoud et al. [65]	20	6	180 game-playing over 12 to 16 recording periods	Motion-Pose Geometric Descriptor (MPGD)
Shim et al. [66]	21	40	Motion capture in four phases (T1–T4) during a reach and grasp task	Melbourne Assessment 2
Povedano et al. [67]	23	16	Eight activities per day with four repetitions per task in a 90 min session	GMFCS, MACS, and SHUEE

**Table 2 sensors-21-07884-t002:** Characteristics of the quantitative measuring devices.

Author	Type of Devices	Manufacturer Device	Parameters	Method	N° 3D Markers	N° of Cameras	Frequency Sampling	Kinematic Model	Algorithm	ISB
Gaillard et al. [51]	OptiTrack optoelectronic system	Motion Analysis, Corvallis, OR, USA	AROM, kinematic analysis	Retroreflective markers	26	12	100 Hz		X Euler angles	*
Fitoussi et al. [55]	Vicon optoelectronic system	Oxford Metrics Group, UK	AROM and PROM, kinematic analysis	Retroreflective markers		6		4 segments	X	
Jaspers et al. [56]	Vicon optoelectronic system	Oxford Metrics Group, UK	Kinematic analysis	Retroreflective markers	17	12–15	100 Hz	5 segments	X BodyMech http://www.bodymech.nl (accessed on 30 August 2021)	*
Howcroft et al. [57]	Vicon optoelectronic system	Oxford Metrics Group, UK	Kinematic analysis	Retroreflective markers	16	7	60 Hz	10 segments	X	
Galli et al. [58]	ELITE2002 optoelectronic system	BTS, Milan, Italy	Kinematic analysis	Retroreflective markers	26		100 Hz		X Euler angles	
Sarcher et al. [59]	Vicon optoelectronic system	Oxford Metrics Group, UK	AROM, kinematic analysis	Retroreflective markers	29	12	100 Hz		X	*
Sevick et al. [60]	Kinect v1	Microsoft, Redmond, WA, USA	AROM	NUI		2	30 Hz		Flexible Action and Articulated Skeleton Toolkit (FAAST) software (Institute for Creative Technologies, CA)	
Anaya Campos et al. [61]	IMU (Shimmer 3^®^)	Shimmer Research, Cambridge, MA, USA	Kinematic analysis, smoothness of movement metrics	Direct via inertial sensor					Spectral Arc Length Metric (SALM), Peaks Metric (PM), Log Dimensionless Jerk Metric (LDJM)	
Mailleux et al. [62]	Vicon optoelectronic system	Oxford Metrics Group, UK	Kinematic analysis	Retroreflective markers	17	12–15	100 Hz	5 segments	Upper Limb Evaluation in Motion Analysis (ULEMA) https://github.com/u0078867/ulema-ul-analyzer (accessed on 30 August 2021)	*
Kim et al. [63]	Accelerometer (Fitmeter)	Fitmeter, Fit.Life Inc., Suwon, Korea	Kinematic analysis	Direct via inertial sensor			128 Hz		Peak acceleration curve, Fitmeter Manager 2 software (Fit.Life Inc., Suwon, Korea)	
Cacioppo et al. [64]	Vicon optoelectronic system	Oxford Metrics Group, UK	AROM, smoothness of movement metrics	Retroreflective markers	26	10	100 Hz		X Arm Profile Score, Spectral arc length (SPARC), Index of curvature	*
Daoud et al. [65]	Kinect v2	Microsoft, Redmond, WA, USA	AROM	NUI		2	30 Hz		X Extended Motion-Pose Geometric Descriptor	
Shim et al. [66]	Vicon optoelectronic system	Oxford Metrics Group, UK	Kinematic analysis	Retroreflective markers			100 Hz		X NEXUS software (Oxford Metrics Group, UK)	
Povedano et al. [67]	Tech-IMU V4	Technaid, Madrid, Spain	Kinematic analysis	Direct via inertial sensor			50 Hz		Tech-MCS V3 System (Technaid, Madrid, Spain)	

ISB: International Society of Biomechanics. X: see reference. * Applied.

**Table 3 sensors-21-07884-t003:** Spatiotemporal parameters of the upper limb.

Device	Movement	ROM (°)	Accuracy	Angular Velocity (°/s)	Acceleration (°/s^2^)	Peak Acceleration (m/s^2^)	PM (%)	SALM (%)	LDJM (%)	SPARC	Timing of Maximal Velocity %	Durations (s)	References
Vicon	Shoulder flexion (+) Extension (−)	−0.22–9.04		−0.39	4.37								[57]
IMU	Hand and wrist						43.74	64.76	33.33				[61]
Vicon	Elbow flexion/extension	54.6–69.3									0.47 +	−0.63 +	[62]
Accelerometer	Elbow flexion/extension					0.80 ± 0.13							[63]
Vicon	Shoulder rotation	52.43								1.67 ± 0.22			[64]
Vicon	Elbow flexion/extension	−0.32 *	−0.25 *										[66]

SPARC: spectral arc length. *: Correlation between the Melbourne assessment (%). +: Correlation between the grip force.

**Table 4 sensors-21-07884-t004:** Reliability, advantages, and disadvantages of the different available instruments for ROM assessments.

References	Instrument/ Device	Manufacturer	ROM	Accuracy	Reliability (ICC)	Advantages	Disadvantages
Universal instrument							
[14]	Goniometer		Passive	5–10°	Intra-rater ± 9.6° Inter-rater ± 8.4°	Inexpensive, transportable, easy to use	Accuracy, its correct use depends mainly on the experience of the evaluator
Optoelectronic device							
[20,21,56]	Vicon	Oxford Metrics Group, UK	Active	<1 mm	Intra-rater: 0.54–0.91 Inter-rater: 0.70–0.96	Accuracy in dynamic and static environments	Retroreflective body markers, expensive
Wearable sensors							
[24,26,27]	IMU	Opal, APDM, Inc., Portland, OR USA	Active	6.8 ± 2.7°	0.930–0.979	Small, portable, wireless, lightweight	Overestimates small joint angles and underestimates large joint angles
Low-cost sensors							
[69]	Kinect v1	Microsoft, Redmond, WA, USA	Active	±5°	0.76–0.98	NUI, low-cost, markerless	Inaccurate measurements in the sagittal plane,
[29,30]	Kinect v2	Microsoft, Redmond, WA, USA	Active	±5°	0.85–0.99 flexion 0.86–0.98 shoulder abduction		inaccurate measurements of ulnar and radial deviations of the upper limbs
Other devices							
[23]	Smartphone applications	Plaincode Software Solutions, Gunzenhausen, Germany	Passive	±3.6°	0.63–0.68	Small, easy to use, affordable	Its correct use depends mainly on the experience of the evaluator

## Data Availability

Not applicable.

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
