# Peer review of "Upper Limb Movement Measurement Systems for Cerebral Palsy: A Systematic Literature Review"

_sensors, 2021, doi:10.3390/s21237884_

Round 1

Reviewer 1 Report

This review identifying and describing available techniques for the quantitative assessment of upper limb active range of motion and kinematics in children with cerebral palsy very systematically. The authors quantified the quality of upper limb movements comprehensively in the background, advantages and disadvantages of different methods. It is suitable for audience. I think it can be published without further modification.

Author Response

Dear reviewer
I hope you are well. Thank you for your comments on this manuscript.
Regards

Reviewer 2 Report

Assessment of upper extremity motor quality is an important and meaningful work, and applying sensor-related technology to this field makes a significant contribution to science. The author summarizes the measurement methods of upper limb movement quality in the last ten years, which has some practical significance. The author's results and discussion are too concise. If the author can significantly improve the quality of the manuscript, it is recommended to accept the publication. In addition, the article has the following problems to be modified. If the author can solve the following problems, it is recommended to accept.

The following are some suggestions to further improve the quality of the paper.

  1. The abstract section, which expands and condenses the content of the abstract.
  2. The Introduction should be improved. In lines 84-93, supplement the development and application related to Kinect camera so that the study contributions could be further presented. For example,“2020, 20, 797. doi: 10.3390/s20030797”,“Remote Sens. 2021, 13, 2288. doi: 10.3390/rs13122288”
  3. In part 2.4 of the text, a total of 14 papers from [49, 53-65] are listed for analysis, and the literature is small and old, supplementing the results of Table 1 analysis.
  4. In section 3.1. ROM assessment, is the advantage of kinect compared to optoelectronic devices only in terms of price? In lines 170-188, add quantitative indicators for comparing performance parameters to better reflect the performance advantages of Kinect.
  5. In sections 4.2 and 4.3, the content of the article is insufficient to draw the conclusion that the future clinical evaluation tool is to use Kinect sensor, which is recommended to be supplemented.
  6. In the conclusion section, the authors are advised to refine it.
  7. Authors should double-check the format of the journal and correct all references to ensure that they are exactly the same as the journal's format.

Author Response

Dear reviewer:

I hope you are well. Thank you for your comments on this manuscript.

Regards

Reviewer 3 Report

This systematic review provides a detail and clear description on the kinemics of a series of devices and their potentials to assessments of AROM.  It's a meaningful work on teh related subjects.

This work pays much attention on kinect. The authors should notice that the MS had terminated the develpment of Kinect; of course, there have been many other products of RGBD cameras, which can be substitions of Kinect.

In another way, technically, an RGBD camera including Kinect may be affected by occlution. So to fuse Kenect with other sensors in motion analysis is nessasary. 

Author Response

(The authors gave the same response as above.)

Reviewer 4 Report

This is an interesting systematic review on the use of sensor technology to measure active range of movement of people suffering with cerebral palsy.

I’ve some comments that I’d like the authors to address and I’ve described them here.

It’s not reasonable to state that “kinematic analysis describes the biomechanical characteristics of motor function and provides objective and accurate information”. The accuracy of the analysis will be affected by the underlying technology used to calculate and measure it. For example, the authors have shown in Table 4 that an IMU sensor has a typical accuracy of 6.8 ± 2.7°, and that both versions of the Kinect sensor have an accuracy of ± 5°.  I recommend that the authors should modify this sentence and to clarify what they mean by this sentence.

The authors describe in lines 52-54 the differences between PROM and AROM, and then they discuss that “AROM of the upper limb is an essential issue in a patient’s rehabilitation process because it provides objective and helpful information in the diagnosis and clinical follow-up of the physical condition progress.” Surely PROM is also an important rehabilitation factor too, particularly with patients who do not have full ROM? Was PROM considered for investigation within the authors PRISMA search? Please explain and justify in more detail why PROM was not considered for this study.

The results shown for the IMU sensor and the smartphone applications shown in Table 4 are interesting. The IMU sensor shows an accuracy of 6.8 ± 2.7° with ICC of 0.930-0.979, and the smartphone applications show an accuracy of ± 3.6 ° with ICC of 0.63-0.68. This is interesting because the underlying technology used to capture movement from both sensors is very similar eg IMU sensors. The authors need to explain in detail what the content of the tables suggest. I would like to see a detailed discussion on the content of each table.

And the VICON system shows an accuracy of <1mm. This accuracy is achievable when retroreflective body markers have some distance between each other, otherwise ghosting and occlusion can occur. And this in turn decreases the accuracy of the VICON measurements. Therefore, VICON is somewhat limited to measurements of larger body segment movement, and would not be suited to smaller movements such as measuring the flexion or extension of MCP and PIP joints of the index and middle fingers as an example.

With these points in mind, I would like to see some more discussions on wearable sensors in section 4.4. I’ve mentioned two points above that I would like to see discussed in more detail in this section.

The authors seem to favour the Kinect sensor without considering the disadvantages of this hardware in more detail. For example, the Kinect sensor is not as portable as other wearable sensors such as an IMU sensor. And the Kinect sensor needs to be set up and secured so that the sensor does not move accidently when it is measuring angular movement. So sensor use is somewhat restricted to a lab environment, or at least to an environment where the sensor can be secured in a fixed position. And the sensor cannot measure movement in 3D, so it is not capable of measuring movement in several planes at any time. I would like to see a more balanced discussion in section 4.5 to account for these disadvantages.

Is the Kinect sensor still on sale? A quick online search shows that common retailers such as Microsoft and Amazon do not have this hardware in stock. How does this affect the authors view on the Kinect sensor?

Author Response

Estimado revisor:

Espero que te encuentres bien. Gracias por sus comentarios sobre este manuscrito.

Consulte el archivo adjunto.

Saludos

Reviewer 5 Report

This manuscript discusses the available techniques for the quantitative assessment of upper limb active range of motion and kinematics in children with cerebral palsy.  The manuscript is written and organized well. 

  1. Firstly, I see a lot of abbreviations through the paper for example World Health Organization (WHO). It occurs once in the manuscript. It is better to remove such less-used abbreviations.    
  2. Besides Kinect, nowadays researchers are using Xsens for human body capturing. The authors have not considered papers related to this. I suggest including a little information/review about this. 
  3. The qualitative study is missing in the paper. 
  4. The discussion and Conclusion section can be extended by discussing future works. 
  5. There are some typo errors. These should be corrected. 

Author Response

(The authors gave the same response as above.)
